Deficient mismatch repair and RAS mutation in colorectal carcinoma patients: a retrospective study in Eastern China

Zhang Xiangyan 1 2
Ran Wenwen 1
Wu Jie 1
Li Hong 1
Liu Huamin 3
Wang Lili 1
Xiao Yujing 1
Wang Xiaonan 1
Li Yujun 1
Xing Xiaoming edithxing@126.com 1
1 Department of Pathology, Affiliated Hospital of Qingdao University , Qingdao , China
2 Department of Pathology, Qingdao University Basic Medicine College , Qingdao , China
3 Department of Oncology, Affiliated Hospital of Qingdao University , Qingdao , China
Coates Philip
Electronic publication date: 2018 Feb 5
Publication date: 2018
Volume: 6
Electronic Location ID: e4341
Received 2017 Oct 12; Accepted 2018 Jan 18
Copyright: ©2018 Zhang et al.
Copyright year: 2018
Copyright holder: Zhang et al.
License: This is an open access article distributed under the terms of the Creative Commons Attribution License, which permits unrestricted use, distribution, reproduction and adaptation in any medium and for any purpose provided that it is properly attributed. For attribution, the original author(s), title, publication source (PeerJ) and either DOI or URL of the article must be cited.
License URL: https://creativecommons.org/licenses/by/4.0/

Keywords: KRAS mutation, Deficient mismatch repair (dMMR ), NRAS mutation, Prognosis, Clinicopathological characteristics, Colorectal carcinoma

Funding: National Natural Science Foundation of China 81201947 Natural Science Foundation of Shandong, China ZR2009CM014 Excellent Young Scientist Foundation of Shandong Province, China 2006BSB14001 Qingdao minsheng science and technology project 17-3-3-38-nsh The present study was supported by grants from the National Natural Science Foundation of China (nos. 81201947), the Natural Science Foundation of Shandong, China (ZR2009CM014), and the Excellent Young Scientist Foundation of Shandong Province, China (no. 2006BSB14001), Qingdao minsheng science and technology project (no.17-3-3-38-nsh). There was no additional external funding received for this study. The funders had no role in study design, data collection and analysis, decision to publish, or preparation of the manuscript.

==============================
Objectives

To investigate the frequency and prognostic role of deficient mismatch repair (dMMR) and RAS mutation in Chinese patients with colorectal carcinoma.

Methods

Clinical and pathological information from 813 patients were reviewed and recorded. Expression of mismatch repair proteins was tested by immunohistochemistry. Mutation analyses for RAS gene were performed by real-time polymerase chain reaction. Correlations of mismatch repair status and RAS mutation status with clinicopathological characteristics and disease survival were determined.

Results

The overall percentage of dMMR was 15.18% (121/797). The proportion of dMMR was higher in patients <50 years old (p < 0.001) and in the right side of the colon (p < 0.001). Deficient mismatch repair was also associated with mucinous production (p < 0.001), poor differentiation (p < 0.001), early tumor stage (p < 0.05) and bowel wall invasion (p < 0.05). The overall RAS mutation rate was 45.88%, including 42.56% (346/813) KRAS mutation and 3.69% (30/813) NRAS mutation (including three patients with mutations in both). KRAS mutation was significantly associated with mucinous production (p < 0.05), tumor stage (p < 0.05) and was higher in non-smokers (p < 0.05) and patients with a family history of colorectal carcinoma (p < 0.05). Overall, 44.63% (54/121) dMMR tumors harbored KRAS mutation, however, dMMR tumors were less likely to have NRAS mutation. Moreover, dMMR, KRAS and NRAS mutation were not prognostic factors for stage I–III colorectal carcinoma.

Conclusions

This study confirms that the status of molecular markers involving mismatch repair status and RAS mutation reflects the specific clinicopathological characteristics of colorectal carcinoma.

Introduction

Colorectal cancer (CRC) is the fourth most common cancer in China, with 331,300 new cases and 159,300 disease-related deaths in 2012 (Chen et al., 2016). The morbidity has increased steadily due to the growth of an aging population and the change of lifestyle in recent years, however, the exact mechanism and related predicted biomarkers are largely unknown.

During the past decades, microsatellite instability (MSI) and RAS mutation have been well studied as two prevalent genetic biomarkers involved in colorectal carcinogenesis. The mismatch repair (MMR) system, which includes the proteins MLH1, MSH2, MSH6 and PMS2, can repair incorrect base-pairing or unmatched DNA loops to maintain genomic stability. MSI is caused by a deficient mismatch repair (dMMR) system, which leads to a high rate of mutations in repeat sequences and accounts for approximately 15% of all CRCs as well as virtually all Lynch syndrome (LS) patients (Geiersbach & Samowitz, 2011; Marra & Boland, 1995; Zhang et al., 2016). Tumors with high level microsatellite instability (MSI-H) caused by germ line mutations or epigenetic silencing of MMR genes have unique clinicopathological characteristics (Cunningham et al., 2010). In early stage CRC, patients with MSI-H demonstrated favorable prognosis compared to those with low level of microsatellite instability (MSI-L) and microsatellite stability (MSS) (Ribic et al., 2003; Sinicrope et al., 2011), however, these patients did not benefit from fluoropyrimidine-based adjuvant chemotherapy (Ribic et al., 2003; Sargent et al., 2010).

The RAS gene family, the other significant biomarker, includes KRAS, NRAS and HRAS, and is located downstream in the epidermal growth factor receptor (EGFR) signal pathway. Mutations in the RAS gene, which are thought to occur early in the adenoma-carcinoma continuum, activate the RAS/MAPK pathway independently of EGFR activation, leading to poor response to EGFR inhibitors (Amado et al., 2008; Punt, Koopman & Vermeulen, 2016). Moreover, National Comprehensive Cancer Network (NCCN) clinical practice guidelines suggested that KRAS and NRAS gene mutations should be detected for metastatic CRC (mCRC) patients before treatment with Cetuximab and Panitumumab (Engstrom et al., 2009).

The status of dMMR and RAS mutation has been widely studied in western countries. The frequency of dMMR CRCs ranged from 15–20% (Giraldez et al., 2010; Sinicrope et al., 2011; Sinicrope et al., 2012), KRAS mutation ranged from 20–50% (De Roock et al., 2010; Naguib et al., 2010; Palomba et al., 2016; Rosty, 2013; Sasaki et al., 2016) and NRAS mutation was noted in less than 5% (De Roock et al., 2010; Palomba et al., 2016; Peeters et al., 2013; Russo et al., 2014). However, studies in China showed a lower frequency of dMMR compared with that in western populations, and the clinicopathological characteristics were also inconsistent (Huang et al., 2010; Jin et al., 2008; Ye et al., 2015). Although several studies reported the frequency of KRAS mutation in Chinese CRC patients, the number of samples was limited in most of these studies (Shen et al., 2011; Ye et al., 2015; Yunxia et al., 2010). Moreover, information about NRAS mutation in Chinese CRC patients was limited. Little has been studied on the association between status of dMMR and RAS mutation. Therefore, in the present study, we analyzed the dMMR and RAS mutation status of CRC patients to evaluate possible associations between dMMR, RAS mutation and the clinicopathological characteristics in primary colorectal carcinoma and we also attempted to explore the prognostic roles of dMMR and RAS mutation.

Materials and Methods

Eight hundred and thirteen formalin-fixed, paraffin-embedded tumor specimens from CRC patients who underwent primary surgical resection from 2013 to 2016 in the Affiliated Hospital of Qingdao University were selected for this study. The patients’ selection method is presented in a consort diagram (Fig. 1). Patients who had undergone preoperative radiotherapy, chemotherapy and/or EGFR-targeted therapy were not included in this study.

Figure 1 Consort diagram in patient selection.

The clinical and pathologic variables were extracted from medical records and pathological reports, which included age, gender, primary locations of tumor, tumor diameter, histological characteristics, TNM stage, smoking status, drinking status and family medication history. The patients were followed up until October 2017, and the data concerning cancer recurrence and patient survival were collected. Patients diagnosed with stage I–III colorectal carcinoma were used to explore the prognostic role of dMMR and RAS mutation with disease-free survival (DFS) and overall survival (OS).

Primary locations of tumors were divided into the right side colon (from the cecum through the transverse colon), the left side colon (from the splenic flexure through the rectosigmoid flexure) and the rectum. Tumors were staged according to the criteria of the seventh edition of the American Joint Commission on Cancer (AJCC) TNM staging system. Mucinous adenocarcinoma and signet-ring cell carcinomas were recorded as mucin-producing tumors.

The study was approved by the Ethics Committee of the Affiliated Hospital of Qingdao University (No.20130049) and all patients had signed informed consent.

Immunohistochemistry for MMR proteins

As previously described (Lin et al., 2014b), all specimens were fixed in 10% neutral buffered formalin and embedded in paraffin blocks. 3 µm-thick tissue sections were used for immunohistochemical analysis. Immunohistochemical staining was performed on an Automated Staining System (BenchMark XT, Ventana Medical Systems, Inc., Tucson, AZ, USA) according to the manufacturer’s instructions. The ready-to-use antibodies were used as follows: MLH1 (No.M1, Ventana Medical Systems Inc, Tucson, AZ, USA, working solution), PMS2 (No.EPR3947, Ventana Medical Systems Inc, Arizona, USA, working solution), MSH2 (No.G219-1129, Ventana Medical Systems Inc, Tucson, AZ, USA, working solution), MSH6 (No.44, Ventana Medical Systems Inc, Tucson, AZ, USA, working solution).

The results were analyzed by two pathologists. Any tumor cell with nuclear staining was recorded as positive staining. Intact expression for all these proteins was regarded as proficient MMR (pMMR). Protein expression was defined as abnormal when nuclear staining of tumor cells was absent in the presence of positive staining in stromal cells and lymphocytes (Fig. 2). The standard criteria for diagnosis of dMMR was as follows: dMMR in MLH1: loss of MLH1 and PMS2; dMMR in MSH2: loss of MSH2 and MSH6; dMMR in MSH6: loss of MSH6; dMMR in PMS2: loss of PMS2 (Richman, 2015).

Figure 2 Immunohistochemical staining for mismatch repair proteins in one case of colorectal carcinoma.

Tumor cells with absent MLH1 (A) and PMS2 (B) expression, and with MSH2 (C) and MSH6 (D) expression, which were regarded as deficient MMR. Note the presence of positive staining in stromal cells and lymphocyte serving as internal positive controls.

Analysis of KRAS and NRAS gene mutations by ARMS-PCR

Formalin-fixed, paraffin-embedded tumor sections were deparaffinized and air dried, and DNA was extracted using the Tiangen Blood and Tissue Kit (TiangenInc, Beijing, China). KRAS (codons12 and 13) and NRAS (codons12, 13 and 61) mutations were detected by amplification refractory mutation system in multiple quantitative polymerase chain reaction (ARMS-multi-qPCR) analysis with the Human KRAS and NRAS Mutation Detection kit (YuanQi Bio-Pharmaceutical Co., Ltd. Shanghai, China). The mutation points detected by this kit are listed in Supplemental Information 2. Codons of RAS were amplified as described previously (Dong et al., 2016). Briefly, 3 µl sample DNA was amplified in a 25 µl reaction containing 9 µl of Mix1 and 13 µl of PCRMix3. Positive and negative controls for each sample were run simultaneously. The program for the PCR amplification flanking KRAS mutation site was as follows: 1 cycle at 42 °C for 5 min; 1 cycle at 94 °C for 3 min; 40 cycles at (94 °C for 15 s; 60 °C for 60 s). Fluorescence signals were collected at 60 °C. The program for the PCR amplification flanking NRAS mutation site was as follows: 1 cycle at 42 °C for 5 min; 1 cycle at 94 °C for 3 min; 40 cycles at (94 °C for 45 s; 60 °C for 80 s). Fluorescence signals were collected at 60 °C. The mutations were identified with a specific probe labeled with Hydroxy fluorescein (FAM). Amplicons were detected using ABI7500 Fast Real-Time PCR System (Thermo Fisher Scientific Inc., Waltham, MA, US).

Statistical analysis

Results were analyzed with SPSS 19.0 (SPSS, Inc, Chicago, IL, USA). For comparison of the frequencies among groups, the Chi-square test and the Fisher exact test were used. Survival curves for DFS and OS were estimated using Kaplan–Meier analysis with the log-rank test. Probability (p) value <0.05 was considered as statistical significance.

Results

Patient characteristics

The main characteristics of the patients are summarized in the Table 1. There were 506 (62.24%) males and 307 (37.76%) females with a mean age of 64 years. The majority of the patients (87.7%) were older than 50 years. 11.69%, 40.84%, 37.15% and 10.33% of patients presented with stage I, stage II, stage III and stage IV disease, respectively. The primary location was more common in rectum (54.49%). There were 283 (34.81%) patients with a smoking history and 165 (20.3%) patients with an alcohol in-taking history, respectively. There were 133 (16.36%) patients with mucin-productive carcinoma.

Table 1 Clinicopathological information of the studied patients (n = 813).

Characteristics	Number	(%)	
Gender			
Male	506	62.24	
Female	307	37.76	
Age			
<50	100	12.3	
≥50	713	87.7	
Location			
Right side colon	181	22.26	
Left side colon	189	23.25	
Rectum	443	54.49	
Mucin production			
With	133	16.36	
Without	680	83.64	
Tumor differentiation			
Poor	138	16.97	
moderate	599	73.68	
Well	33	4.06	
Unknown	43	5.29	
Tumor stage			
I	95	11.69	
II	332	40.84	
III	302	37.15	
IV	84	10.33	
Bowel wall invasion (T)			
T1	21	2.58	
T2	104	12.79	
T3	336	41.33	
T4	352	43.3	
Lymph node metastasis (N)			
N0	458	56.33	
N1	203	24.97	
N2	152	18.7	
Distant metastasis (M)			
M0	729	89.67	
M1	84	10.33	
Lymphovascular invasion			
Yes	339	41.7	
No	462	56.83	
Unknown	12	1.47	
Alcohol intake			
Ever	165	20.3	
Never	648	79.7	
Smoking			
Ever	283	34.81	
Never	530	65.19	
Colorectal family history			
Yes	48	5.9	
No	337	41.45	
Unknown	428	52.65	

MMR status and associations with clinicopathological characteristics

MMR status was successfully evaluated in 797 patients. 121 (15.18%) patients exhibited dMMR. The rates of dMMR deficiency in MLH1, PMS2, MSH2 and MSH6 were 9.78% (78/797), 1.25% (10/797), 3.26% (26/797) and 0.87% (7/797), respectively. The rates of deficiency in MLH1/PMS2 and MSH2/MSH6 were 11.92% (88/797) and 4.14% (33/797), respectively. The association of clinicopathological characteristics with MMR status is presented in Table 2. The proportion of dMMR was higher in patients <50 years old (p < 0.001). A higher rate of dMMR was found in stage II cancers (19.02%, p = 0.019). dMMR status was also associated with mucinous production (p < 0.001), poor differentiation (p < 0.001) and localization of the tumor to the right side of the colon (p < 0.001). dMMR patients had a higher propensity to bowel wall invasion (p = 0.018).

Table 2 Correlations between mismatch repair protein deficiency and clinicopathological characteristics (n = 797).

Characteristics	Number	dMMR	MLH1/ PMS2	MSH2/MSH6	
		Defective (%)	P value	Defective (%)	P value	Defective (%)	P value	
Gender								
Male	495	73 (14.75)	0.662	52 (10.51)	0.561	21 (4.24)	0.853	
Female	302	48 (15.89)		36 (11.92)		12 (3.97)		
Age								
<50	99	29 (29.29)	<0.001	23 (23.23)	<0.001	6 (6.06)	0.284*	
≥50	698	92 (13.18)		65 (9.31)		27 (3.87)		
Location								
Right side colon	173	61(35.26)	<0.001	43 (24.86)	<0.001	18 (10.4)	<0.001	
Left side colon	185	25 (13.51)		18 (9.73)		7 (3.78)		
Rectum	439	35 (7.97)		27 (6.15)		8 (1.82)		
Mucin production								
With	131	36 (27.48)	<0.001	25 (19.08)	<0.001	11 (8.4)	0.007	
Without	666	85 (12.76)		63 (9.46)		22 (3.3)		
Tumor differentiation								
Poor	134	36 (26.87)	<0.001	24 (17.91)	<0.001	12 (8.96)	0.012*	
Moderate	589	71 (12.05)		51(8.66)		20 (3.39)		
Well	31	4 (12.9)		3 (9.68)		1 (3.23)		
Unknown	43							
Tumor stage								
I	94	6 (6.38)	0.019	5 (5.32)	0.110	1 (1.06)	0.288*	
II	326	62 (19.02)		45 (13.81)		17 (5.21)		
III	301	41 (13.62)		30 (9.97)		11 (3.65)		
IV	76	12 (15.79)		8 (10.52)		4 (5.26)		
Bowel wall invasion (T)								
T1	20	3 (15)	0.018	2 (10)	0.139	1 (5)	0.067*	
T2	102	5 (4.9)		5 (4.9)		0 (0)		
T3	334	59 (17.66)		44 (13.17)		15 (4.49)		
T4	341	54 (15.83)		37 (10.85)		17 (4.98)		
Lymph node metastasis (N)								
N0	445	74 (16.63)	0.192	54 (12.13)	0.354	20 (4.49)	0.583	
N1	200	31 (15.5)		22 (11)		9 (4.5)		
N2	152	16 (10.53)		12 (7.89)		4 (2.63)		
Distant metastasis (M)								
M0	721	110 (15.26)	0.550	80 (12.13)	0.88	30 (4.16)	0.929*	
M1	76	11 (14.47)		8 (10.53)		3 (3.95)		
Lymphovascular invasion								
Yes	335	47 (14.03)	0.451	35 (10.45)	0.679	12 (3.58)	0.481	
No	457	73 (15.97)		52 (11.38)		21 (4.59)		
Unknown	5							
Alcohol intake								
Ever	162	19 (11.72)	0.170	13 (8.02)	0.170	6 (3.7)	0.755	
Never	635	102 (16.06)		75 (11.81)		27 (4.25)		
Smoking								
Ever	263	35 (13.31)	0.170	24 (9.13)	0.226	11 (4.18)	0.967	
Never	534	86 (16.1)		64 (11.98)		22 (4.12)		
Colorectal family history								
Yes	48	11 (22.92)	0.071	5 (10.42)	0.795	6 (12.5)	0.016*	
No	335	44 (13.13)		32 (9.55)		12 (3.58)		
Unknown	414							
Notes.

* Fisher’s exact test was used.

Although dMMR tumors were present more often in patients with CRC family history, no significant difference (22.92% vs 13.13%, p > 0.05) was found in this study. The loss of MSH2/MSH6 expression was more often observed in patients with CRC family history (12.5% vs 3.58%, p = 0.016). In other respects, the patients with tumors exhibiting dMMR were similar to those exhibiting pMMR.

RAS gene mutation and associations with clinicopathological characteristics

RAS status was tested from 813 patients. The mutation rates of KRAS and NRAS were 42.56% (346/813) and 3.69% (30/813), respectively. There were three patients demonstrating mutation in both KRAS and NRAS. Patients suffering from tumors with mucinous production had a higher incidence of KRAS mutation compared with those having tumors without mucinous production (54.89% vs 40.18%, p = 0.002). A higher rate of KRAS mutation was found in stage II (48.49%) compared with that in stage I, stage III and stage IV (36.84%, 40.45%, 34.52%, respectively) cancers (p = 0.023) and in non-smokers compared with smokers (46.6% vs34.98%, p = 0.001). Patients with CRC family history also showed higher rate of KRAS mutation (54.17% vs 37.39%, p = 0.013). Tumors with RAS mutation showed lower propensity to lymph node metastasis (p = 0.006) and distant metastasis (p = 0.048). No significant associations between KRAS mutation and other clinicopathological characteristics were found in the present study. Meanwhile, NRAS mutation was not significantly associated with any clinicopathological characteristics (Table 3).

Correlations between RAS mutation and MMR status

RAS mutation rate was slightly higher in pMMR tumors than in dMMR tumors, but failed to reach a significant difference (46.3% vs 44.63%, p > 0.05). There was also no obvious correlation between MMR status and KRAS mutation (42.3% vs 44.63%, p > 0.05). No NRAS mutation was detected in dMMR tumors. Compared with dMMR tumors, pMMR tumors had a higher propensity to harbor NRAS mutation (p = 0.009, Table 4). The distribution of MMR and KRAS status is shown in Supplemental Information 3. Correlation between KRAS gene mutation and clinicopathological characteristics in dMMR tumors is summarized in Table 5. No significant association between KRAS mutation and any clinicopathological characteristics were found in dMMR tumors.

Prognostic value of dMMR and RAS mutation in stage I–III CRC

Of the 813 followed-up patients, 729 patients were diagnosed with stage I–III CRC, including 95 stage I patients, 332 stage II patients and 302 stage III patients. dMMR and RAS mutation were not prognostic for DFS and OS in stage I–III CRC (Fig. 3). Of the 121 dMMR patients, 109 patients were diagnosed with stage I–III CRC and 45.87% (50/109) patients harbored KRAS mutation. However, KRAS mutation was not prognostic factor for these patients (Fig. 4).

Table 3 Correlations between RAS gene mutations and clinicopathological characteristics (n = 813).

Characteristics	Number	RAS	KRAS	NRAS	
		Mutation (%)	P value	Mutation (%)	P value	Mutation (%)	P value	
Gender								
Male	506	221 (43.68)	0.105	204 (40.32)	0.097	19 (3.75)	0.9	
Female	307	152 (49.51)		142 (46.25)		11 (3.58)		
Age								
<50	100	38 (38)	0.091	37 (37)	0.23	1 (1)	0.161*	
≥50	713	335 (46.98)		309 (43.34)		29 (4.07)		
Location								
Right side colon	181	91 (50.28)	0.178	88 (48.62)	0.097	3 (1.66)	0.164	
Left side colon	189	77 (40.74)		71 (37.57)		6 (3.17)		
Rectum	443	205 (46.28)		187 (42.21)		21 (4.74)		
Mucin production								
With	133	74 (55.64)	0.014	73 (54.89)	0.002	1 (0.75)	0.087	
Without	680	299 (43.97)		273 (40.18)		29 (4.22)		
Tumor differentiation								
Poor	138	55 (39.86)	0.315	54 (39.13)	0.604	1 (0.72)	0.093	
Moderate	599	276 (46.08)		251 (41.9)		28 (4.67)		
Well	33	17 (51.52)		16 (48.48)		1 (3.03)		
Unknown	43							
Tumor stage								
I	95	41 (43.16)	0.031	35 (36.84)	0.023	6 (6.32)	0.18*	
II	332	170 (51.2)		161 (48.49)		9 (2.71)		
III	302	133 (44.04)		122 (40.4)		14 (4.64)		
IV	84	29 (34.52)		28 (34.52)		1 (1.19)		
Bowel wall invasion (T)								
T1	21	9 (42.86)	0.36	8 (38.1)	0.158	1 (4.76)	0.36*	
T2	104	40 (38.46)		34 (32.69)		6 (5.77)		
T3	336	154 (45.83)		146 (43.45)		9 (2.68)		
T4	352	170 (48.3)		158 (44.89)		14 (3.98)		
Lymph node metastasis (N)								
N0	458	224 (48.91)	0.006	209 (45.63)	0.079	15 (3.28)	0.265	
N1	203	88 (43.35)		83 (40.89)		6 (2.96)		
N2	152	61 (40.13)		54 (35.53)		9 (5.92)		
Distant metastasis (M)								
M0	729	343 (47.05)	0.048	317 (43.48)	0.116	29 (3.98)	0.353*	
M1	84	30 (35.71)		29 (34.52)		1 (1.19)		
Lymphovascular invasion								
Yes	339	157 (46.31)	0.763	145 (42.77)	0.825	14 (4.13)	0.623	
No	462	209 (45.24)		194 (41.99)		16 (3.46)		
Unknown	12							
Alcohol intake								
Ever	165	67 (40.61)	0.128	63 (38.18)	0.203	5 (3.03)	0.615	
Never	648	306 (47.22)		283 (43.67)		25 (3.86)		
Smoking								
Ever	283	109 (38.52)	0.002	99 (34.98)	0.001	10 (3.53)	0.863	
Never	530	264 (49.81)		247 (46.6)		20 (3.77)		
Colorectal family history								
Yes	48	28 (58.33)	0.017	26 (54.17)	0.013	3 (6.25)	0.178*	
No	337	135 (40.95)		126 (37.39)		9 (2.67)		
Unknown	428							
Notes.

* Fisher’s exact test was used.

Table 4 Correlations between mismatch repair protein deficiency and RAS status (n = 797).

MMR status	RAS	KRAS	NRAS	
	Mutant/tested cases (%)	P value	Mutant/tested cases (%)	P value	Mutant/tested cases (%)	P value	
dMMR	54/121 (44.63)	0.734	54/121 (44.63)	0.635	0/121 (0)	0.009*	
MHL1/PMS2 deficiency	39/88 (44.32)	0.725	39/88 (44.32)	0.875	0/88 (0)	0.044*	
MSH2/MSH6 deficiency	15/33 (45.45)	0.999	15/33 (45.45)	0.72	0/33 (0)	0.391*	
pMMR	313/676 (46.3)		286/676 (42.3)		30/676 (4.43)		
Notes.

* Fisher’s exact test was used.

Table 5 Correlations between KRAS gene mutations and clinicopathological characteristics in dMMR tumors (n = 121).

Characteristics	Number	KRAS	P value	
		Mutation (%)		
Gender				
Male	73	31 (42.47)	0.555	
Female	48	23 (47.91)		
Age				
<50	29	11 (37.93)	0.405	
≥50	92	43 (46.74)		
Location				
Right side colon	61	26 (42.62)	0.891	
Left side colon	25	12 (48)		
Rectum	35	16 (45.71)		
Mucin production				
With	36	20 (55.56)	0.116	
Without	85	34 (40)		
Tumor differentiation				
Poor	36	10 (27.78)	0.099*	
Moderate	71	35 (49.3)		
Well	4	2 (50)		
Unknown	10			
Tumor stage				
I	6	2 (33.33)	0.277*	
II	62	33 (53.2)		
III	41	15 (36.59)		
IV	12	4 (33.33)		
Bowel wall invasion (T)				
T1	3	2 (66.67)	0.179*	
T2	5	0 (0)		
T3	59	26 (44.07)		
T4	54	26 (48.15)		
Lymph node metastasis (N)				
N0	74	38 (51.35)	0.056	
N1	31	13 (41.94)		
N2	16	3 (18.75)		
Distant metastasis (M)				
M0	110	50 (45.45)	0.753*	
M1	11	4 (36.36)		
Lymphovascular invasion				
Yes	47	21 (44.68)	0.927	
No	73	32 (43.83)		
Unknown	1			
Alcohol intake				
Ever	19	9 (47.37)	0.855	
Never	102	46 (45.1)		
Smoking				
Ever	35	14 (40)	0.514	
Never	86	40 (46.51)		
Colorectal family history				
Yes	11	5 (45.45)	0.589	
No	44	24 (54.55)		
Unknown	66			
Notes.

* Fisher’s exact test was used.

Figure 3 Survival curves for disease free survival (DFS) and overall survival (OS) in stage I–III colorectal carcinoma according to dMMR or RAS status.

(A) Disease free survival (DFS) according to dMMR status; (B) overall survival (OS) according to dMMR status; (C) DFS according to KRAS status; (D) OS according to KRAS status; (E) DFS according to NRAS status; (F) OS according to NRAS status.

Figure 4 Survival curves for disease free survival (DFS) and overall survival (OS) in stage I–III dMMR colorectal carcinoma according to KRAS status.

(A) Disease free survival (DFS) according to KRAS status; (B) overall survival (OS) according to KRAS status.

Discussion

As prognostic and predictive biomarkers, MMR deficiency and RAS mutation are important for clinical treatment and prognosis of CRC patients. Compared with pMMR, patients with dMMR CRCs are reported to have unique clinicopathological characteristics such as poor differentiation, early stage, increased tumor-infiltrating lymphocytes and better clinical outcome (Brenner, Kloor & Pox, 2014; Korphaisarn et al., 2015; Ribic et al., 2003). The RAS gene is a predictive biomarker for the resistance to anti-EGFR monoclonal antibody (MoAb) treatment in mCRCs (Amado et al., 2008; Punt, Koopman & Vermeulen, 2016). However, geographic and racial differences between Chinese and other countries were reported (Huang et al., 2010; Ismael et al., 2017; Kim et al., 2007; Vasovcak et al., 2011; Ye et al., 2015), which need to be validated with large sample amounts. Furthermore, data regarding RAS mutation frequency and dMMR CRC is not consistent in China. Thus, we designed this study in the Chinese population aiming to explore the relationship between the RAS mutation, MMR status and clinicopathological parameters, also expecting to find some prognostic and predictive biomarkers for CRC.

Our results demonstrated an overall MMR deficiency rate of 15.18%, which is within the established range of 15–21% (Giraldez et al., 2010; Sinicrope et al., 2012; Carethers et al., 2004; Cushman-Vokoun et al., 2013), but slightly higher than that reported from other Chinese populations (Huang et al., 2010; Jin et al., 2008; Ye et al., 2015). Reports from Korea (Jung et al., 2012) and Japan (Kadowaki et al., 2015) which used PCR-based MSI testing also showed that the frequencies of MSI-H CRCs were around 10%. This discrepancy can be explained by the different detective methods to some extent. Compared with PCR-based MSI testing examination, immunohistochemistry is thought to be easily available and time-saving. Furthermore, immunohistochemistry may detect MMR-deficient cases that can be potentially missed by PCR-based MSI testing (Shia, 2008).

Correlations between dMMR status and clinicopathological characteristics were controversial (Ismael et al., 2017; Jin et al., 2008; Ribic et al., 2003; Sinicrope et al., 2011). Reports from three independent Chinese groups (Huang et al., 2010; Jin et al., 2008; Ye et al., 2015) indicated that dMMR had specific associations such as female gender, right sided colon tumors and mucious tumors. In a study including 1,063 CRCs, Lin et al. (2014a) observed that MSI was associated not only with gender, tumor location and mucin production, but also with tumor differentiation and tumor stage. In our current study, we found patients younger than 50 tended to be dMMR. These diverse findings may be attributed to different criteria for age division, ethnicities, environmental factors as well as the specificity and sensitivity of the detection methods.

In our study, there was a correlation between MSH2/MSH6 deficiency and family history of CRC, but not MLH1/PMS2 deficiency. In addition, according to the Bethesda criteria (Burt et al., 2010), 12 CRCs were diagnosed with LS. In MSH2/MSH6 deficient CRCs, 33.3% (6/18) were LS, while in MLH1/PMS2 defective cases, 13.95% (6/43) were LS, suggesting MSH2/MSH6 deficient patients had higher opportunity to be diagnosed with LS. Some of the recent studies may help to explain this finding: the majority dMMR CRCs were caused by inactivation of MLH1 and more than 70% MLH1 deficiency was caused by MLH1 promoter hypermethylation (Hampel et al., 2005), which could distinguish sporadic dMMR CRCs from LS cases, therefore, most MLH1 defective tumors were sporadic CRC. Another interesting phenomenon in our investigation is that we found most patients’ family medical history was unclear and they did not know whether other family members had polyps removed, moreover, many cancers might be prevented by early stage colonoscopy, so the family history may be deceptive (Hampel, 2014). Therefore, screening strategy based on family history may be improper. All patients with newly diagnosed CRC should be screened for LS (Hampel, 2014). Inconsistent with previous studies, which indicated that patients with dMMR tumors had significantly better survival than that of pMMR patients (Des Guetz et al., 2009; Korphaisarn et al., 2015; Lanza et al., 2006), our study showed that dMMR was not a prognostic factor for patients with stage I–III CRC, although the incidence of dMMR in stage III disease was lower, suggesting that dMMR tumors had lower propensity to metastasize.

In the present study, the mutation rates of KRAS and NRAS are 42.56% and 3.69%, respectively. The KRAS mutation rate is significantly higher than the value of 20.7% among 314 CRC patients from Taiwan, China (Liou et al., 2011), 22% among 202 CRC patients from the England (Naguib et al., 2010), 30.1% among 392 CRC patients from Switzerland (Zlobec et al., 2010), but similar to that previously reported in Guangzhou, China (43.9%, 25/57) (Mao et al., 2012). Several factors may lead to such differences, such as sample size, dietary and lifestyle factors, as well as racial and/or environmental differences. Furthermore, we detected the coding sequence of codon12 and codon13 in exon 2 of the KRAS gene, which may help to explain the higher percentage of KRAS mutation than those detected in codon12 only. Except for exon 2, recent studies have shown 5–10% of tumors harbored exon 3 or exon 4 mutation (Janakiraman et al., 2010; Lin et al., 2014a), which would also result in resistance to anti-EGFR inhibitors. Therefore, extending the detection spectrum of RAS might help to optimize the selection of the CRC patients to receive anti-EGFR MoAbs.

The frequency of KRAS mutation has been reported to be associated with age, gender, differentiation and tumor stage (Gao et al., 2012; Li et al., 2011; Ye et al., 2015; Yunxia et al., 2010; Zhu et al., 2012). Inconsistent with these results, our study showed that KRAS mutation was associated with mucin production, tumor stage, non-smoking and CRC family history. RAS mutated tumors showed lower propensity to lymph node and distant metastasis. No convincing evidence demonstrates that KRAS mutation is an independent prognostic factor for CRC (Jin et al., 2008; Palomba et al., 2016; Russo et al., 2014; Yunxia et al., 2010). In the present study, no associations of KRAS mutation with DFS and OS were found in patients with stage I–III CRC. Further studies based on longer follow-up time and larger sample size are needed to confirm this conclusion.

In our study, the percentage of the four tumor subgroups, including dMMR/KRAS mutation, dMMR/KRAS wild-type, pMMR/KRAS mutation and pMMR/KRAS wild-type tumors was 6.78%, 8.4%, 35.88%, 48.94%, respectively, which is similar to the data reported by a study from Beijing, China (Ye et al., 2015). According to recent reports (Nash et al., 2009; Roth et al., 2010), patients with a MSS/KRAS mutant tumor had the worst survival than the other three groups. Therefore, dMMR and KRAS markers may provide a foundation for developing a molecular prognostic scoring system for CRC patients in the future.

Previous studies have shown that pMMR patients tended to harbor more KRAS mutation than dMMR patients (Naguib et al., 2010; Ye et al., 2015). One hypothesis for this result is that BRAF and KRAS mutations were almost mutually exclusive in CRC and MSI tumors are more likely to harbor a BRAF mutation, so MSS tumors might harbor more KRAS mutations (Naguib et al., 2010). However, in the present study, we did not find any differences in KRAS mutation between pMMR and dMMR tumors, and further studies based on larger sample size are needed to explore this controversy in Chinese CRCs.

Additionally, our study provided an opportunity to investigate the status of KRAS mutation in Chinese dMMR patients. KRAS mutation presented in 44.63% dMMR patients in our study, similar to previous studies in western countries (Cushman-Vokoun et al., 2013; Oliveira, 2004). All of these results indicate that KRAS mutation could be quite common in dMMR tumors. There were no associations between KRAS mutation and clinicopathologic characteristics in dMMR tumors. A study conducted by Nash et al. (2009) indicated that KRAS status was an independent prognostic factor in early stage MSI CRC patients. Moreover, MSI patients with wild-type KRAS and BRAF tumors have more favorable prognosis than patients with mutated KRAS or BRAF tumors in early stage CRC (De Cuba et al., 2016; Phipps et al., 2015). However, we did not find KRAS mutation as a prognostic factor for dMMR patients with stage I–III CRC.

NRAS, as one of the RAS family, showed close relations with KRAS. Unlike KRAS, NRAS mutation was rarely detected in CRC patients. In our study, the mutation rate of NRAS was 3.69%, similar to previous reports (Chang et al., 2016; Irahara et al., 2010; Palomba et al., 2016; Peeters et al., 2013). Moreover, we observed 25/388 KRAS wild-type tumors with NRAS mutation, which can partially help to explain the resistance to anti-EGFR MoAb in some KRAS wild-type patients. Considering the heavy financial burden in MoAb treatment in CRC patients, NRAS mutation should be tested before MoAb treatment in KRAS wild-type tumors. Another interesting phenomenon is that no NRAS mutation was detected in dMMR patients, which suggested NRAS mutation might be mutually exclusive with dMMR. Meanwhile, NRAS mutation was not significantly associated with any clinicopathologic characteristics in our study.

However, our results should be elucidated with consideration of its limitations: first, the sample size was relatively small, rendering some findings inconclusive; second, we used a commercially available kit authenticated by China Food and Drug Administration (CFDA) and the mutation subgroups were uncertain. A study conducted by Lin et al. (2014a) demonstrated that mutation in KRAS codon12 was associated with significantly poorer outcome than mutations elsewhere or wild-type KRAS. Therefore, the subgroup of mutation codons should be carefully explored in future; third, we did not collect data of clinical management, therefore, the influence of clinical treatment for survival was uncertain.

Conclusion

In conclusion, this was an exploratory analysis of correlations between RAS mutation and MMR status with clinicopathological characteristics in Eastern Chinese CRC patients. The status of these molecular markers, involving MLH1/PMS2, MSH2/MSH6, KRAS and NRAS mutation, reflects the specific clinicopathological characteristics of CRC. More comprehensive molecular classification and survival analysis should be explored in future experiments.

Supplemental Information

Supplemental Information 1 Raw data of 813 cases

Click here for additional data file.

Supplemental Information 2 The mutation points that detected by our kit named the Human KRAS and NRAS Mutation Detection kit

We listed the mutation points that detected by our kit named the Human KRAS and NRAS Mutation Detection kit (YuanQi Bio-Pharmaceutical Co., Ltd. Shanghai, China).

Click here for additional data file.

Supplemental Information 3 The distribution of mismatch repair protein and KRAS mutation in colorectal cancer patients (n = 797)

The distribution of mismatch repair protein and KRAS mutation in colorectal cancer patients.

Click here for additional data file.

Additional Information and Declarations

Competing Interests

Author Contributions

Human Ethics

Data Availability

The authors declare there are no competing interests.

Xiangyan Zhang conceived and designed the experiments, performed the experiments, wrote the paper.

Wenwen Ran conceived and designed the experiments, performed the experiments, contributed reagents/materials/analysis tools, wrote the paper.

Jie Wu performed the experiments, reviewed drafts of the paper.

Hong Li contributed reagents/materials/analysis tools.

Huamin Liu reviewed drafts of the paper.

Lili Wang analyzed the data.

Yujing Xiao prepared figures and/or tables.

Xiaonan Wang conceived and designed the experiments, performed the experiments, prepared figures and/or tables.

Yujun Li analyzed the data, prepared figures and/or tables.

Xiaoming Xing conceived and designed the experiments, contributed reagents/materials/analysis tools, wrote the paper, reviewed drafts of the paper.

The following information was supplied relating to ethical approvals (i.e., approving body and any reference numbers):

The study was approved by the Ethics Committee of the Affiliated Hospital of Qingdao University. Approval number: No.QDFY-20130049.

The following information was supplied regarding data availability:

The raw data has been provided in the Supplemental Files.

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
