# Peer review of "Deficient mismatch repair and RAS mutation in colorectal carcinoma patients: a retrospective study in Eastern China"

_PeerJ, doi:10.7717/peerj.4341_

## Round 0.1 · original submission · Major Revisions

Both reviewers have raised a number of issues, mainly relating to your definitions of patient characteristics for RAS and MMR status, but also relating to the assays you have used.

If you decide to submit a revision, you will need to address all of the issues raised by the two reviewers within the revised manuscript (not only in a letter to the editor).

Please also check the manuscript for language - PeerJ does not perform language editing during Production.

Due to the large number of issues that have been raised and their importance, it is highly likely that the revised manuscript will be re-reviewed before a final decision is reached.

Reviewer 1 ·

Basic reporting

In this study 'Deficient mismatch repair and RAS mutation in colorectal carcinoma patient: a retrospective study in Eastern China', the authors Zhang et al. conducted a well-designed clinical study basing on 813 Chinese patients with colorectal carcionma, and investigated the correlations between the clinicopathological characteristics and surival of patients with their dMMR status or RAS mutations. The overall design of the study is well-laid out, with clear background information given in the introduction part. The discussion part is very nicely written, which sumarized a range of related and updated studies in China and western countries in this field. Moreover, the authors commented on the limitation of their studies, which does help improve the integrity of the whole paper.

The English language is generally well-used, however there are several places in the text where the language/grammer should be improved, for example lines 48, 112, 146, 155-156, 229, 232, 297, 311. In addition, abbreviation should be spelled out at its first appreance. In line 179, the full name of pMMR should be indicated.

Experimental design

The research question in the study is well defined, the authors also nicely stated the knowledge gap this study will fill in. The experiments used are appropriate, however, there are several issues need to be further addressed to support the authors' conclusion.

1. The authors replied on IHC to determine the presence/deficiency of the different MMR proteins. However, there is no definition as of how the authors determine what level of staining indicates protein expression deficiency. The authors should show/describe their standard of definition, positive and negative controls, and show representative IHC images of patients with either pMMR or dMMR.

2. As the authors stated in the discussion, the RAS mutations they detected rely on a kit produced in China. The mutations that could be picked up by this kit could be different from those reported in studies based on western populations. The authors should, therefore, list the mutations that are detected by this method, in order to improve reproductivity of the study and information sharing in this field.

Validity of the findings

The authors did a great job in performing statistical analysis, the conclusions are generally supported by robust data. However, as the authors discussed, the small sample size in some studies (i.e. survival analysis) may lead to insignificant results. The authors should include power analysis at the beginning of these comparisons, to test how many subjects are needed to detect a statistically significant difference, and if their sample size has reached this power. This will help further determine the validity of the study/results.

One minor point is that in the statistical analysis section, the authors should add a description as of how they correct type I and II errors to adjust their results.

Additional comments

Nice study with important implication for finding biomarkers for colorectal cancer patients in China. There are certain points that could be further improved to enhance the validity of the results and the flow of the paper.

Reviewer 2 ·

Basic reporting

The manuscript “Deficient mismatch repair and RAS mutation in colorectal carcinoma patients: a retrospective study in Eastern China” By Zhang st al investigated the frequency of KRAS mutation and deficient MMR in colorectal cancer patients which report the frequency of 45.88% and 15.43%, respectively. In addition, dMMR and KRAS mt did not appear as prognostic factors in this population.
Overall, this is a clear and concise manuscript. The introduction is relevant and theory based. The strength of this study is a relatively large sample size (N=813) to detect the mutation frequencies. The methods are generally appropriate. However, there are several papers which reported the frequency of RAS and dMMR in Chinese population and little data is added from this manuscript. Moreover, the definition of dMMR should be revised to the standard criteria. Finally, this manuscript has several grammatical errors and needs to be improved before publishing.

Experimental design

no comment

Validity of the findings

no comment

Additional comments

Major:
Material and Methods
1. You excluded the patients who received preoperative EGFR-targeted therapy (which mean RAS wt patients) that lead to selection bias in this cohort. So, the prevalence of the KRAS gene mutation in your study might be higher than normal.
2. Please explain the reason why you did not include all of the patients that had tissue available for mutation testing in that period?
3. Please explain the reason why you had the survival data in only 15.6% (114/729) in the early stage of the disease?
4. Also, Might be better if you demonstrate the patient selection method in consort diagram.
5. Line 116-117, “Loss of expression of any MMR protein was regards as dMMR”.
Actually you have to use the standard criteria for diagnosis of dMMR (https://www.ncbi.nlm.nih.gov/pmc/articles/PMC4583524/ ) which are
dMMR in MLH1: loss of expression in MLH1 and PMS2
dMMR in MSH2: loss of expression in MSH2 and MSH6
dMMR in MSH6: loss of expression in MSH6
dMMR in PMS2: loss of expression in PMS2

From your definition, if the IHC result show loss of expression in MLH1, you will interpret as dMMR which is incorrect.



Result
1. You should categorize the patients into 4 groups according to the standard definition for dMMR. From Table 2, there is a huge confuse? difference in the number. For example, female gender had dMMR in 50 cases. However, when you define in MLH-1/PMS2=42 cases and MSH2/MSH6=12, total of 53 cases.
Discussion
1. Line 207: the MMR deficiency rate should be changed with the standard definition



Minor:

1. Should use the term of “right/left sided colon rather than right/left colon” in all parts of the manuscript
2. Line 55: favorable prognosis and could not benefit of adjuvant 5FU based chemotherapy in which stage of the disease? You should specific only in early stage.
Line 65: not only KRAS gene but the NCCN recommend both KRAS and NRAS gene mutation
Line 166: RAS status was "tested" from 813 patients…… not "detected"
Line 172: should be 46.6% vs 34.98%, P=0.001
Line 220: specific the association i.e female gender, right sided tumor… etc

---

## Round 0.2 · Minor Revisions

Many thanks for submitting your revised manuscript, which will be suitable for publication in PeerJ.

I am recommending that minor revisions be made to improve the language and clarify some sentences. I have performed some light editing of the manuscript to achieve this for you (see the pdf attached).
When revising your manuscript, please refer to the pdf and alter your text accordingly, or indicate in your response letter why you have not accepted these language/grammar corrections.

In addition:

Figure 1: please change text to;
Diagnosed as CRC (N=1,522)
Refused to participate (N=501)

Figure 3: please change the legend to;
stage I - III

Please change the x axis for all six graphs to;
Time (months)

---

## Round 0.3 · accepted · Accept

Many thanks for revising the manuscript. I have no further concerns.